# Global-Local Graph Neural Networks for Node-Classification

**Moshe Eliasof**
Computer Science Department,
Ben-Gurion University of the Negev, Israel
eliasof@post.bgu.ac.il

**Eran Treister**
Computer Science Department,
Ben-Gurion University of the Negev, Israel
erant@cs.bgu.ac.il

## Abstract

The task of graph node-classification is often approached using a *local* Graph Neural Network (GNN), that learns only local information from the node input features and their adjacency. In this paper we propose to benefit from global and local information through the form of learning *label-* and *node-* features to improve node-classification accuracy. We therefore call our method Global-Local-GNN (GLGNN). To learn proper label features, for each label, we maximize the similarity between its features and nodes features that belong to the label, while maximizing the distance between nodes that do not belong to the considered label. We then use the learnt label features to predict the node-classification map. We demonstrate our GLGNN using GCN and GAT as GNN backbones, and show that our GLGNN approach improves baseline performance on the node-classification task.

## 1 Introduction

The field of Graph Neural Networks (GNNs) has gained large popularity in recent years [1–5] in a wide variety of fields and applications such as computer graphics and vision [5–9], Bioinformatics [10, 11], node-classification [3, 12, 13] and others. In the context of node-classification, most of the methods consider only nodal (i.e., local) information by performing local aggregations and $1 \times 1$ convolutions, e.g., [3, 12–14]. In this paper we propose to incorporate label (i.e., global) information to improve the training of GNNs. In particular, we propose to learn a feature vector for each label (class) in the data, which is then used to determine the final prediction map and is mutually utilized with the learnt node features. Because our method is based on learning global features that scale as the number of labels in the dataset, our method does not add significant computational overhead compared to the backbone GNNs. We show the generality of this approach by demonstrating it on GCN [3] and GAT [12] on a variety of node-classification datasets, both in semi- and fully-supervised settings. Our experiments reveal that our GLGNN approach is beneficial for all the considered datasets, and we also illustrate the learnt global features with respect to the node features for a qualitative assessment of our method. Our contributions are as follows:

- We propose to learn *label* features to capture global information of the input graph.
- We fuse label- and node- features to predict a node-classification map.
- We demonstrate our method qualitatively by illustrating the learnt label features in Fig. 1 and quantitatively by demonstrating the benefit of using GLGNN approach on 6 real-world datasets.

## 2 Related Work

### 2.1 Graph Neural Networks

Typically, graph neural networks (GNNs) are categorized into spectral [1] and spatial [3, 5, 15–17] types. While the former learns a global convolution kernel, it scales as the number of nodes in the graph, $n$, and is of a higher computational complexity. To obtain local convolutions, spatial GNNs formulate a local-aggregation scheme is usually implemented using the Message-Passing Neural

M. Eliasof et al., Global-Local Graph Neural Networks for Node-Classification (Extended Abstract). Presented at the First Learning on Graphs Conference (LoG 2022), Virtual Event, December 9–12, 2022.

Network mechanism [17], where each node aggregates features (messages) from its neighbours, according to some policy. In this work we follow the latter, whilst adding a global mechanism by learning label features to improve accuracy on node-classification tasks.

## 2.2 Improved training of GNNs

To improve accuracy performance, recent works introduce new training policies, objective functions and augmentations. A common trick for training on small datasets like Cora, Citeseer and Pubmed is the incorporation of Dropout [18] after every GNN layer, which has become a standard practice [3, 13, 14, 19]. Other methods suggest to randomly alternate the data rather than the GNN neural units. For example, DropEdge [20] and DropNode [21] randomly drop graph edges and nodes, respectively. In the work PairNorm [22], the authors propose a normalization layer that alleviate the over-smoothing phenomenon in GNNs [23]. Another approach is the Mixup technique that enriches the learning data, and has shown success in image classification [24, 25]. Following that, the work GraphMix [26] proposed an interpolation-based regularization method by parameter sharing of GNNs and point-wise convolutions and [27] proposed a graph mixup policy based on graph topology and subgraphs.

Other methods consider the training of GNNs from an information and entropy point of view following the success of mutual information in CNNs [28]. For example, DGI [29] learns a global graph vector and considers its correspondence with local patch vectors. However, it does not consider label features as in our work. In the work InfoGraph [30] the authors learn a discriminative network for graph classification tasks, and in [31] a consistency-diversity augmentation is proposed via an entropy perspective for node and graph classification tasks.

## 3  Notations

We denote an undirected graph by the tuple $\mathcal{G} = (\mathcal{V}, \mathcal{E})$ where $\mathcal{V}$ is a set of $n$ nodes and $\mathcal{E}$ is a set of $m$ edges, and by $\mathbf{f}^{(l)} \in \mathbb{R}^{n \times c}$ the feature tensor of the nodes $\mathcal{V}$ with $c$ channels at the $l$-th layer. The adjacency matrix is defined by $\mathbf{A} \in \mathbb{R}^{n \times n}$, where $\mathbf{A}_{ij} = 1$ if there exists an edge $(i, j) \in \mathcal{E}$ and 0 otherwise, and the diagonal degree matrix is denoted $\mathbf{D}$ where $\mathbf{D}_{ii}$ is the degree of the $i$-th node.

Let us also denote the adjacency and degree matrices with added self-edges by $\tilde{\mathbf{A}}$ and $\tilde{\mathbf{D}}$, respectively. Using this notation, for example, the propagation operator from GCN [3] is obtained by $\tilde{\mathbf{P}} = \tilde{\mathbf{D}}^{-\frac{1}{2}} \tilde{\mathbf{A}} \tilde{\mathbf{D}}^{-\frac{1}{2}}$, and its architecture is given by

$$\mathbf{f}^{(l+1)} = \text{ReLU}(\tilde{\mathbf{P}} \mathbf{f}^{(l)} \mathbf{K}^{(l)}), \tag{1}$$

where $\mathbf{K}^{(l)}$ is a $1 \times 1$ convolution matrix.

We consider the node-classification task with $k$ labels. We denote the ground-truth labels by $\mathbf{y} \in \mathbb{R}^{n \times k}$ and the node-classification prediction by applying SoftMax to the output of the network $\mathbf{f}^{out}$

$$\hat{\mathbf{y}} = \text{SoftMax}(\mathbf{f}^{out}) \in \mathbb{R}^{n \times k}. \tag{2}$$

## 4  Method

We now describe the local and global feature extraction mechanism and our objective functions.

**Local features.**  The local information is obtained by learning node features $\mathbf{f} \in \mathbb{R}^{n \times d}$ using some backbone denoted by GNN. In our experiments, we evaluate our method with GNN being a GCN [3] as in Eq. (1) or a GAT [12]. Note that our GLGNN approach does not assume a specific GNN backbone and thus can possibly be utilized with other GNNs.

**Global features.**  Our global information mechanism learns label features $\mathbf{g} \in \mathbb{R}^{k \times d}$. Specifically, to obtain the global features we consider the concatenation of initial nodes-embedding $\mathbf{f}^{(0)}$ and the last GNN layer node features $\mathbf{f}^{(L)}$ denoted by $\left[\mathbf{f}^{(0)} \oplus \mathbf{f}^{(L)}\right]$. We then perform a single $1 \times 1$ convolution denoted by $\mathbf{K}_{\text{g}}$, followed by a ReLU activation, and feed it to a global MaxPool readout function to obtain a single vector $\mathbf{s} \in \mathbb{R}^d$. Formally:

$$\mathbf{s} = \text{MaxPool}\left(\text{ReLU}\left(\mathbf{K}_{\text{g}}\left[\mathbf{f}^{(0)} \oplus \mathbf{f}^{(L)}\right]\right)\right). \tag{3}$$

Using the global vector $\mathbf{s}$, we utilize $k$ (the number of labels) multi-layer perceptrons (MLPs) that are implemented as an inverted bottleneck [32], and in particular resembles the squeeze-and-excite mechanism from [33]. Each MLP is comprised of the following:

$$\mathbf{g_i} = \mathbf{K_s}\left(\mathrm{ReLU}\left(\mathbf{K}_e\mathbf{s}\right)\right), \tag{4}$$

where $\mathbf{K}_e, \mathbf{K}_s$ are an expanding (from $d$ to $e \times d$) and shrinking (from $e \times d$ to $d$) $1 \times 1$ convolutions, and the expansion rate $e$ is a hyper-parameter which is set $e = 12$ in our experiments. Note that this operation can be efficiently implemented using a grouped convolution to obtain $\mathbf{g} = [\mathbf{g}_0, \ldots, \mathbf{g_{k-1}}]$ in parallel. Also, because $\mathbf{s}$ is a vector, the computational overhead is rather low compared to the total complexity of the backbone GNN.

**Node-classification map.** To obtain a node-classification prediction map, we consider matrix-vector product of the final GNN output $\mathbf{f}^{(L)} \in \mathbb{R}^{n \times d}$ with each of the label features $\mathbf{g}_i \in \mathbb{R}^d$ in (4). More formally for each label we obtain the following node-label correspondence vector:

$$\mathbf{z}_i = \mathbf{f}^{(L)} \cdot \mathbf{g}_i \in \mathbb{R}^n. \tag{5}$$

By concatenating the $k$ correspondence vectors and applying the SoftMax function, we obtain a node-classification map

$$\hat{\mathbf{y}} = \mathrm{SoftMax}\left(\mathbf{z}_0 \oplus \ldots \oplus \mathbf{z_{k-1}}\right) \in \mathbb{R}^{n \times k}, \tag{6}$$

which is the final output of our GLGNN.

**Objective functions.** To train our GLGNN we propose to minimize the following objective function:

$$\mathcal{L} = \mathcal{L}_{CE} + \alpha\mathcal{L}_{\mathrm{GL}}, \tag{7}$$

where $\mathcal{L}_{CE}$ denotes the cross-entropy loss between ground-truth $\mathbf{y}$ and predicted node labels $\hat{\mathbf{y}}$ from Eq. (6). $\alpha$ is a positive hyper-parameter, and $\mathcal{L}_{\mathrm{GL}}$ denotes a global-local loss that considers the relationship between the label and node features by demanding the similarity of nodes that belong to a respective label while requiring the dis-similarity of node features that do not belong to that label and its features, as follows

$$\mathcal{L}_{\mathrm{GL}} = \sum_{l=0}^{k-1}\left(\sum_{\mathbf{y}_i=l}\|\mathbf{g}_l - \mathbf{f}_i^{(L)}\|_2^2 - \sum_{\mathbf{y}_i \neq l}\min\left(\|\mathbf{g_l} - \mathbf{f_i^{(L)}}\|_2^2, \mathrm{r}\right)\right), \tag{8}$$

where $\min(\cdot, \cdot)$ is a clamping function that returns the minimal values of its arguments, and $\mathrm{r}$ is a positive hyper-parameter, as is standard with contrastive losses [34]. In our experiments we set $\mathrm{r} = 10$.

## 5 Experiments

We now demonstrate GLGNN on semi- and fully-supervised node-classification. Our GLGNN consists of an embedding layer ($1 \times 1$ convolution), a series of GNN backbone layers and the label features MLPs as described in Sec. 4. As GNN backbones, we consider GCN [3] and GAT [12]. We elaborate on the specific architecture in Appendix A. We use the Adam [35] optimizer, and perform a grid-search to choose the hyper-parameters (see Appendix B for more information). Our code is implemented using PyTorch [36] and PyTorch-Geometric [37], trained on an Nvidia Titan RTX GPU.

We show that for all the considered tasks and datasets, our GLGNN offers a consistent improvement over the baseline methods, and besides the obtained accuracy we report the relative accuracy improvement compared to the baseline GCN and GAT methods. Also, we find that our GLGNN is competitive with recent state-of-the-art methods. We provide further datasets information in Appendix C.

### 5.1 Semi-Supervised Node-Classification

We consider Cora, Citeseer and Pubmed [38] datasets and their standard, public training/validation/testing split as in [39], with 20 nodes per class for training. We follow the training and evaluation scheme of [13] and consider various GNN models like GCN, GAT, superGAT [40], APPNP [41], JKNet [42], GCNII [13], GRAND [43], PDE-GCN [44], pathGCN [45], EGNN[14] and superGAT [40]. We also consider other improved training techniques P-reg [46], GraphMix [26] and NASA [31]. We summarize the results in Tab. 1 and illustrate the learnt labels and nodes features in Fig. 1, revealing the clustering effect of learning label nodes.

**Table 1:** Semi-supervised node-classification accuracy (%).

| Method | Cora | Citeseer | Pubmed |
|---|---|---|---|
| GCN | 81.1 | 70.8 | 79.0 |
| GAT | 83.1 | 70.8 | 78.5 |
| APPNP | 83.3 | 71.8 | 80.1 |
| JKNET | 81.1 | 69.8 | 78.1 |
| GCNII | 85.5 | 73.4 | 80.3 |
| GRAND | 84.7 | 73.6 | 81.0 |
| PDE-GCN | 84.3 | 75.6 | 80.6 |
| pathGCN | 85.8 | 75.8 | 82.7 |
| EGNN | 85.7 | – | 80.1 |
| superGAT | 84.3 | 72.6 | 81.7 |
| GraphMix | 84.0 | 74.7 | 81.1 |
| P-reg | 83.9 | 74.8 | 80.1 |
| NASA | 85.1 | 75.5 | 80.2 |
| GLGCN (ours) | 84.2 $_{(+3.8\%)}$ | 73.3 $_{(+3.5\%)}$ | 81.5 $_{(+3.1\%)}$ |
| GLGAT (ours) | 84.5 $_{(+1.6\%)}$ | 72.6 $_{(+2.5\%)}$ | 81.2 $_{(+3.4\%)}$ |

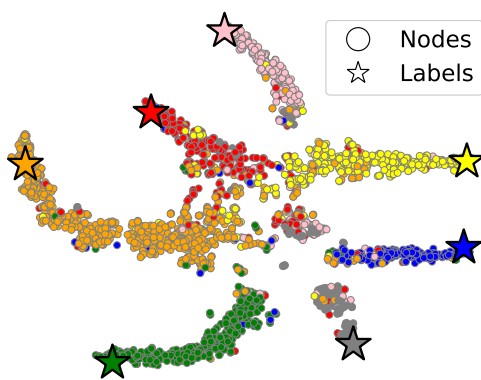

**Figure 1:** tSNE embedding of learnt label- and node-features of Cora. The similarity of the label features and the corresponding node features shows the clustering effect of incorporating global information.

**Table 2:** Fully-supervised node-classification accuracy (%) on *homophilic* datasets.

| Method | Cora | Citeseer | Pubmed |
|---|---|---|---|
| Homophily | 0.81 | 0.80 | 0.74 |
| GCN | 85.77 | 73.68 | 88.13 |
| GAT | 86.37 | 74.32 | 87.62 |
| Geom-GCN | 85.27 | 77.99 | 90.05 |
| APPNP | 87.87 | 76.53 | 89.40 |
| JKNet (Drop) | 87.46 | 75.96 | 89.45 |
| GCNII | 88.49 | 77.08 | 89.57 |
| WRGAT | 88.20 | 76.81 | 88.52 |
| GCNII* | 88.01 | 77.13 | 90.30 |
| GGCN | 87.95 | 77.14 | 89.15 |
| H2GCN | 87.87 | 77.11 | 89.49 |
| GLGCN (ours) | 88.47 $_{(+3.1\%)}$ | 77.72 $_{(+5.4\%)}$ | 88.61 $_{(+0.05\%)}$ |
| GLGAT (ours) | 88.65 $_{(+2.6\%)}$ | 77.37 $_{(+4.1\%)}$ | 88.74 $_{(+0.1\%)}$ |

**Table 3:** Fully-supervised node-classification accuracy (%) on *heterophilic* datasets.

| Method | Corn. | Texas | Wisc. |
|---|---|---|---|
| Homophily | 0.30 | 0.11 | 0.21 |
| GCN | 52.70 | 52.16 | 48.92 |
| GAT | 54.32 | 58.38 | 49.41 |
| Geom-GCN | 60.81 | 67.57 | 64.12 |
| JKNet (Drop) | 61.08 | 57.30 | 50.59 |
| GCNII | 74.86 | 69.46 | 74.12 |
| GCNII* | 76.49 | 77.84 | 81.57 |
| GRAND | 82.16 | 75.68 | 79.41 |
| WRGAT | 81.62 | 83.62 | 86.98 |
| GGCN | 85.68 | 84.86 | 86.86 |
| H2GCN | 82.70 | 84.86 | 87.65 |
| GraphCON-GCN | 84.30 | 85.40 | 87.80 |
| GraphCON-GAT | 83.20 | 82.20 | 85.70 |
| GLGCN (ours) | 74.86 $_{(+42.0\%)}$ | 70.27 $_{(+34.7\%)}$ | 65.29 $_{(+33.4\%)}$ |
| GLGAT (ours) | 75.67 $_{(+39.3\%)}$ | 70.01 $_{(+19.9\%)}$ | 65.88 $_{(+33.3\%)}$ |

## 5.2 Fully-Supervised Node-Classification

To further validate the efficacy of our method, we employ fully-supervised node-classification on 6 datasets, namely, Cora, Citeseer, Pubmed, Cornell, Texas and Wisconsin using the 10 random splits from [47] with train/validation/test split of $48\%, 32\%, 20\%$ respectively, and report their average accuracy. In all experiments, we use 64 channels and perform a grid-search to determine the hyper-parameters. We compare our accuracy with methods like GCN, GAT, Geom-GCN [47], APPNP, JKNet [42], WRGAT [48], GCNII [13], DropEdge [20], H2GCN [49], GGCN [50] and GraphCON [51]. We distinguish between homophilic and heterophilic datasets, and report the results of the former in Tab. 2, and of the latter in Tab. 3, where we also report the homophily score of each dataset (adapted from [47]). We see an improvement across all benchmarks and types of datasets compared to the baseline methods of GCN and GAT and competitive results on homophilic datasets with recent state-of-the-art methods.

## 6 Conclusion

In this paper we propose GLGNN, a method to leverage global information for semi- and fully-supervised node-classification. By learning and fusing global label features and local node features, we show that it is possible to cluster the nodes in a way that enables improved classification accuracy and demonstrate that our method outperforms baseline models by a significant margin. Future research directions include the evaluation of this method on graph classification datasets and exploring additional possible methods of global label information extraction and incorporation.

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

# A    Architecture

We now elaborate on the specific architecture used in our experiments in Sec. 5. Our network architecture consists of an opening (embedding) layer ($1 \times 1$ convolution), a sequence of GNN backbones layers (see details below for specific aggregation rules for GCN and GAT), and a series of $1 \times 1$ convolutions to learn the global labels features. We have a single type of architecture, based on the scheme of GCN [3] for node-classification. The difference between our GLGCN and GLGAT is the backbone of the GNN. We specify the node feature extraction architecture in Tab. 4, and the label feature extraction architecture in Tab. 5. In what follows, we denote by $c_{in}$ and $k$ the input and output channels, respectively, and $c$ denotes the number of features in hidden layers. We initialize the embedding and label features related layers with the Glorot [52] initialization, and $\mathbf{K}^{(l)}$ from Eq. (1) is initialized with an identity matrix of shape $c \times c$. We denote the number of GNN layers by $L$, and the dropout probability by $p$.

The GCN [3] backbone is given by:

$$\mathbf{f}^{(l+1)} = \mathrm{ReLU}(\tilde{\mathbf{P}}\mathbf{f}^{(l)}\mathbf{K}^{(l)}). \tag{9}$$

as described in Eq. (1) in the main text

**GAT.** The GAT [12] backbone defines the propagation operator:

$$\alpha_{ij}^{(l)} = \frac{\exp\left(\mathrm{LeakyReLU}\left(\mathbf{a}^{(l)\top}[\tilde{\mathbf{K}}^{(l)}\mathbf{f}_i^{(l)} \oplus \tilde{\mathbf{K}}^{(l)}\mathbf{f}_j^{(l)}]\right)\right)}{\sum_{p \in \mathcal{N}_i} \exp\left(\mathrm{LeakyReLU}\left(\mathbf{a}^{(l)\top}[\tilde{\mathbf{K}}^{(l)}\mathbf{f}_i^{(l)} \oplus \tilde{\mathbf{K}}^{(l)}\mathbf{f}_p^{(l)}]\right)\right)}, \tag{10}$$

where $\mathbf{a}^{(l)} \in \mathbb{R}^{2c}$ and $\tilde{\mathbf{K}}^{(l)} \in \mathbb{R}^{c \times c}$ are trainable parameters and $\oplus$ denotes channel-wise concatenation and the neighbourhood of the $i$-th node is denoted by $\mathcal{N}_i = \{j | (i,j) \in \mathcal{E}\}$.

By gathering all $\alpha_{ij}^{(l)}$ for every edge $(i,j) \in \mathcal{E}$ into a propagation matrix $\mathbf{S} \in \mathbb{R}^{n \times n}$ we obtain the GAT architecture:

$$\mathbf{f}^{(l+1)} = \mathrm{ReLU}(\mathbf{S}^{(l)}\mathbf{f}^{(l)}\mathbf{K}^{(l)}). \tag{11}$$

**Table 4:** The architecture used for node features extraction.

| Input size | Layer | Output size |
|---|---|---|
| $n \times c_{in}$ | Dropout(p) | $n \times c_{in}$ |
| $n \times c_{in}$ | $1 \times 1$ Convolution | $n \times c$ |
| $n \times c$ | ReLU | $n \times c$ |
| $n \times c$ | $L\times$ GNN backbone | $n \times c$ |

**Table 5:** The architecture used for label features extraction. The input of this architecture is the output of Tab. 4

| Input size | Layer | Output size |
|---|---|---|
| $n \times c$ | MaxPool | $1 \times c$ |
| $1 \times c$ | $k \times 1 \times 1$ Convolutions | $k \times 12 \cdot c$ |
| $k \times 12 \cdot c$ | ReLU | $k \times 12 \cdot c$ |
| $k \times 12 \cdot c$ | $k \times 1 \times 1$ Convolutions | $k \times c$ |

# B    Hyper-parameters

We perform a grid-search to determine the hyper-parameters values. In Tab. 6 we specify each hyper parameter and the range of values that we considered.

# C    Datasets

The statistics of the datasets used in our experiments are provided in Tab. 7.

**Table 6:** Hyper-parameters and considered range for grid-search. LR and WD denote the learning rate and weight decay of embedding and label feature extraction layers. $LR_{GNN}$ and $WD_{GNN}$ denote the learning rate and weight decay of the GNN layers. $\alpha$ is the balancing coefficient from Eq. (7).

| Hyper-parameter | Values range |
|---|---|
| LR | [1e-1, 1e-2, 1e-3, 1e-4] |
| $LR_{GNN}$ | [1e-1, 1e-2, 1e-3, 1e-4] |
| WD | [1e-3, 1e-4, 1e-5, 0] |
| $WD_{GNN}$ | [1e-3, 1e-4, 1e-5, 0] |
| $\alpha$ | [1e+2, 1e+1,1, 1e-1,1e-2] |
| $p$ | [0.5,0.6,0.7] |

**Table 7:** Datasets statistics.

| Dataset | Classes | Nodes | Edges | Features |
|---|---|---|---|---|
| Cora | 7 | 2,708 | 5,429 | 1,433 |
| Citeseer | 6 | 3,327 | 4,732 | 3,703 |
| Pubmed | 3 | 19,717 | 44,338 | 500 |
| Cornell | 5 | 183 | 295 | 1,703 |
| Texas | 5 | 183 | 309 | 1,703 |
| Wisconsin | 5 | 251 | 499 | 1,703 |

