# OpenReview forum: "Global-Local Graph Neural Networks for Node-Classification"
_logconference.io/LOG/2022/Conference — LoG 2022 Poster_

### Official Review · Reviewer_ZcnF · 2022-10-19

**Overall Score:** 3
**Confidence:** 4

**Review:**

**Summary**

The paper introduces a graph representation learning method GLGNN which uses global information via label features and local information via node features. Specifically, GLGNN is trained to maximize the distance between nodes with different labels. GLGNN improves node classification accuracy compared to basic GNNs such as GCN and GAT but shows lower accuracy compared to state-of-the-art methods.

**Strength**

- The paper is well written with clear motivation, to use both global and local information.
- The proposed method does not add significant computational overhead compared to the backbone GNNs.
- The method can be used for general GNNs, improving the accuracy of the backbone.

**Weakness**

- The experimental results do not show any advantage over state-of-the-art methods. The results on heterophilic datasets are not comparable to the state-of-the-art models, with over a 15pp accuracy gap. The proposed method should be improved on the heterophilic setting to claim that using global and local information is beneficial for graph representation learning. Note that comparing the training/evaluation time could show the advantage of the method.
- Further, the accuracy results need statistical tests such as t-test to show the significance of the improvements.
- Analysis of the benefit of using both the global and local information, for example as an ablation study is necessary, as it is the key idea of the method.
- There is no explanation about the result of Figure 1, the tSNE embedding.

**Recommendation**

I clearly reject this paper, as the paper has more room to improve. The concerns I have addressed in the **Weakness** should be addressed.

---

### Official Review · Reviewer_TmEY · 2022-10-21

**Overall Score:** 5
**Confidence:** 4

**Review:**

The paper proposes a method to incorporate global and local information in the graph. The so-called global information is primarily the feature information of the node. The paper provides a way to gather global features corresponding to the labels in the graphs using a readout function on the convoluted first and last-layer local representation of all nodes, which then gives the feature vector corresponding to each label using inverse bottleneck implemented with the help of MLPs. The objective function is to minimize the cross entropy and the global local loss, which minimizes the difference in global and local representations of the labels.

The method is clearly described. The experiments show that the model improves by a significant margin over baseline, however, they do not show significant improvements over the existing methods for node classification in a semi-supervised and fully supervised setting.

---

### Official Review · Reviewer_quXB · 2022-10-22

**Overall Score:** 5
**Confidence:** 4

**Review:**

This paper aims to incorporate global information in graph neural networks to further improve the accuracy in node classification tasks. Specifically, the authors design a feature vector for each label and then adopt it for the final prediction. A global-local loss is also proposed to learn a good label features by maximizing the intra similarity and inter dissimilarity between the label features and GNN output. The designed GLGNN is model-agnostic and can be easily plugged in the representative GNNs. Experiments demonstrate that GLGNN can achieve higher accuracy than many representative baselines.


Strengths and weakness:
(S1) This paper is well-written and easy to follow. The contributions of this paper are clear to me.

(S2) The idea of incorporating global and local features in GNN is interesting and make sense to me. The authors pay many efforts to illustrate model pipe design, especially for global features.

(W1) Even though incorporating global information in GNN is interesting, the motivation and why incorporating global information in this way are unclear. For example, are there any preliminary experiments or previous works showing why global information matters in GNN?  As for model design, what’s the design rationale behind the proposed pipeline? Why consider the concatenation of initial and the last layer of embedding? What’s the advantage of resembling the squeeze-and-excite mechanism? More justifications should be provided.

(W2) The experiments are not convincing. (a) The authors claim that the proposed method does not add significant computation overhead. However, there is not any computation complexity analysis and experiment results to support this claim. Please provide the computation complexity of this method and baselines. The training or inference time of the proposed method should be provided compared with several representative backbones. (b) The experimental dataset is toy and not representative. OGB dataset, Reddit, especially for large-scale GNN benchmark, are the standard dataset in node classification tasks. It would be better to conduct experiments on these datasets. (c) Ablation study, and hyperparameter studies are missing in the current version. (d) Experimental results show the proposed method is not SOTA. For example, in table 1, the accuracy of the proposed method in Cora and Citeseer dataset is lower than pathGCN. For fully-supervised node-classification accuracy on homophilic and heterophilic datasets, the proposed method is not also the best.

(W3) There are many missing related works. For example, in Mixup augmentation, there are many other mixup data augmentation methods, such as if-Mixup [1], G-mixup [2], Graph-Corp [3], Graph Mixup [4]. In Related work section, the authors mention many line of research in GNNs, many of which are actually not that related. I suggest the authors mainly focus on GNNs backbone design.

Based on the above-mentioned weakness, I vote weak reject for the current version. I would like to improve the score if the authors can tackle my major concerns. Looking forward to the authors’ feedback.


[1] Hongyu Guo and Yongyi Mao. ifmixup: Towards intrusion-free graph mixup for graph classification. arXiv, 2110, 2021.

[2] Xiaotian Han, Zhimeng Jiang, Ninghao Liu, and Xia Hu. G-mixup: Graph data augmentation for graph classification. arXiv preprint
arXiv:2202.07179, 2022.

[3] Yiwei Wang, Wei Wang, Yuxuan Liang, Yujun Cai, and Bryan Hooi. Graphcrop: Subgraph cropping for graph classification. arXiv preprint
arXiv:2009.10564, 2020.

[4] Yiwei Wang, Wei Wang, Yuxuan Liang, Yujun Cai, and Bryan Hooi. Mixup for node and graph classification. In Proceedings of the Web Conference 2021, pages 3663–3674, 2021.

---

### Official Review · Reviewer_f6tG · 2022-10-27

**Overall Score:** 5
**Confidence:** 4

**Review:**

Summarize the contributions of this work:

This paper deals with node classification using graph neural networks. To improve the performance of GNNs, the authors propose GLGNN to leverage both local features and global features to obtain the node classification map and optimize the learning model, where local features are node features and global features are label features. Experiments on the public datasets are conducted to evaluate the effectiveness of GLGNN.

Strong points:
1. Labeled-augmented or enhanced graph neural networks have been widely investigated, while this paper provides a new formulation to take advantage of label information.
2. GLGNN learns label features as global features from node features through an inverted bottleneck mechanism, which seems interesting.

Weak points:
1. Though this is a shot paper, some model learning steps are unclear without many justifications. For example, why the inverted bottleneck works for label feature learning from a single vector s obtained by global-pooling node features? Also, why the clamping function in L_GL can enforce dissimilarity of node features with r=10? For my understanding, contrastive loss among label features g, similar and dissimilar node features l seems a better choice for global-local loss.
2. GLGNN outperforms GCN and GAT, but compared to other improved GNN models (Table 1-3), GLCNN only delivers comparable or worse performance. Also, there are some other label-augmented graph neural networks, which are overlooked for the comparisons in this paper.
3. There are some confusions raised by the matrix dimensions. For example, in Eq. (3), the concatenation of f0 and fL is nx(d+c); to perform global max-pooling, Kg seems to be (d+c)xd, which should be placed after [f0+fL] instead of before. Also, in Eq. (5), fL is the output of the last GNN layer, whose dimension is nxc, while the dimension of gi is d; in this case, how the product can be performed here?

---

### Meta-Review · Area_Chair_rcxi · 2022-11-16

**Confidence:** 3
**Recommendation:** Borderline and needs further discussi…

**Meta Review:**

This paper proposes a novel algorithm for gathering global and local information for node classification. The reviewers generally thought the method was clear and where there were issues the discussion seemed to sort things out well. However, there is uniform concern about the performance of the method, which can be applied to other graph models that don't incorporate global information, in terms of more recent node classification baselines. The authors response that the value of their work is not on achieving SOTA but on demonstrating the usefulness of their algorithm is valid, and I think especially for a short paper this is appropriate. However, I'm still not sure if some of the concerns, e.g. wanting complexity in the paper, wanting to see this method demonstrated on top of stronger baselines that don't use global information justifies rejection.

---

### Decision · Program_Chairs · 2022-11-23

Accept (Poster)